# Fano-Like Resonance of Heat-Reconfigurable Silicon Grating Metasurface Tuned by Laser-Induced Graphene

**DOI:** 10.3390/nano13030492

**Published:** 2023-01-25

**Authors:** Yukuan Ma, Yulei Huang, Yuehong Zhu, Hao Zhou, Congliao Yan, Shutong Wang, Guoliang Deng, Shouhuan Zhou

**Affiliations:** 1College of Electronics and Information Engineering, Sichuan University, Chengdu 610064, China; 2North China Research Institute of Elector-Optics, Beijing 100015, China

**Keywords:** metasurface, Fano resonance, plasmon, laser-induced graphene

## Abstract

We propose a heat-reconfigurable metasurface composed of the silicon-based gold grating. The asymmetric Fano-like line shape is formed due to the mutual coupling of the local surface plasmon (LSP) in the gap between the two layers of Au gratings and the surface propagating plasmon (SPP) on the surface of the Au gratings. Then, we effectively regulate the Fano resonance by applying a bias voltage to laser-induced graphene (LIG), to generate Joule heat, so that the resonant dip of one mode of the Fano resonance can shift up to 28.5 nm. In contrast, the resonant dip of the other mode barely changes. This effectively regulates the coupling between two resonant modes in Fano resonance. Our study presents a simple and efficient method for regulating Fano-like interference in the near-infrared band.

## 1. Introduction

Fano resonance is a kind of asymmetric resonance different from the traditional Lorentz spectral line shape. It originates from the quantum interference phenomenon caused by the interaction between the continuous state and the discrete state in atomic physics [1,2,3]. A large number of nano-metal patterns have been verified to achieve effective Fano resonance [4,5,6,7,8,9,10,11,12]. The one-dimensional grating structure plays an important role in the metasurface. Compared with other complex metasurfaces, the structure is simpler. The one-dimensional grating can provide research on interesting physical phenomena, such as plasmon lattice [13,14,15,16]. In addition, a one-dimensional metal grating structure was proposed to form an apparent Fano resonance-like linear shape through the coupling interference between surface propagating plasmon (SPP) and local surface plasmon (LSP) [17,18,19]. The LSP often has a large bandwidth due to its large radiation damping [20]. In contrast, the SPP is a narrow-band electromagnetic mode, which can be effectively coupled to interfere with the original LSP to form Fano resonance. In these reports, excellent refractive index sensing is achieved through Fano interference in a one-dimensional gold grating made on a transparent medium substrate. However, it is necessary to regulate Fano resonance dynamically. In optical switches, the resonance wavelength and intensity can also be modulated. On the other hand, if the sensors using plasmon resonance can be tuned, the detection window can be adjusted, and the signal-to-noise ratio can be increased to improve sensitivity [21]. In this paper, we designed the metasurface of the silicon grating and, then, coated the silicon grating with a gold film to form the upper and lower layers of the Au gratings. The coupling between the excited SPP mode and LSP mode, on the two-layer Au gratings, forms a spectral line of Fano-like interference. At the same time, the influence of parameter of the grating on the Fano resonance is studied, and it is found that this kind of composite grating has better tunable performance on the line shape of the Fano resonance. The Fano resonance can be tuned by changing its grating parameters. We then applied different voltages to the LIG at the bottom of the device to dynamically tune the Fano resonance and found that the Joule heat accumulated in silicon only effectively tuned the excitation mode as a continuous state. In contrast, the mode as a discrete state was hardly affected. This means we can dynamically tune the coupling of the two modes in Fano resonance through voltage. In addition, the device can realize Fano resonance only by preparing a one-dimensional grating without complex metal patterns, which greatly reduces the preparation cost and has potential in optical communication, nonlinear optics, and optical sensing.

## 2. Structure and Method

We primarily employed electron beam lithography (EBL), reactive ion etching (RIE), and vacuum thermal evaporation during the preparation phase. First, the structural parameters of the device are designed, and then, a customized pattern is formed on the silicon wafer coated with photo-resist using electron beam lithography technology. Then, a silicon grating is formed using reactive ion etching technology. Finally, a gold film with a thickness of 80 nm is coated on the upper and lower surfaces of the silicon grating by using vacuum coating technology. The structural diagram and images of the scanning electron microscope (SEM) are shown in Figure 1, where the period p of the prepared grating is 1100 nm, the width w of the grating is 952 nm, and the height h1 of the grating is 120 nm.

First, we simulate the prepared devices with the finite element method. The incident port is polarized along the *x*-axis in the electric field direction. The boundary condition in the x-horizontal direction is periodic, while the boundary condition in the z-direction is the perfect absorption layer (PML) layer. The refractive index of silicon is 3.47. The refractive index data of gold is from the test data in the literature [22]. We also tested the reflection spectrum of the device. The schematic diagram of the experimental apparatus is shown in Figure 2. After passing through the beam collimator, the supercontinuum laser is vertically concentrated on the structure’s surface after being focused by the objective lens. The reflected spectrum is collected by the objective lens, moved to the optical fiber, and analyzed by the spectrometer.

It is found that the polarization angle determines the resonant peak of its reflection spectrum. When the incident light is polarized in the x direction, we can observe resonant dips that form interference, as shown in Figure 3. We used the parameters above to conduct a finite element simulation, as shown in Figure 3, and it can be found that the simulation is in good agreement with the measured result.

## 3. Discussion

### 3.1. Analysis of Fano Resonance Formation

The extinction ratio of Mode 1 is 90%, the resonance wavelength is 1170 nm, and the corresponding full width at half maximum (FWHM) is 46 nm. The resonant wavelength of Mode 2 is 1450 nm, the corresponding FWHM is 102 nm, and the absorption is close to 50%. To study the physical mechanism of the formation of asymmetric line types in the reflection spectrum, we simulated the electric field distribution of Mode 1 and Mode 2 in Figure 4a,b. Its electric field mode shows the characteristics of typical SPP, in which the field energy at the interface between the medium and the metal is greatly limited to the metal surface, and shows a fast attenuation pattern in its normal direction [23]. We can further see another typical feature of SPP mode from the magnetic field of Mode 1, as shown in Figure 4c. The electric field energy of Mode 2 is mainly concentrated in the corner of the two-layer Au grating, which shows the typical characteristics of the electric dipole radiation of LSP. Meanwhile, there is also SPP 3 mode with weak electric field intensity propagating on the lower surface of the upper layer of Au gratings. The electric field intensity is far less than that of SPP 1 and SPP 2, as shown in Figure 4b. The asymmetric line type between Mode 1 and Mode 2 is formed by the mutual coupling of SPP mode excited by the Au grating, and LSP mode in Mode 2. The two-interference cancellation forms a Fano-like asymmetric resonance curve. The LSP mode at the corners of the two-layer gratings in Mode 2 obviously acts as a continuous state with a wide spectrum. This also explains why the FHWM of Mode 2 is almost twice that of Mode 1. Therefore, the Fano resonance is formed after SPP mode is coupled with the LSP mode.

### 3.2. Analysis of the Influence of Composite Au Grating Parameters on Fano Resonance

First, we analyzed the period p of the grating by scanning the period width, within the range of 1100 nm–1400 nm, while keeping the parameters of other devices unchanged. The two-dimensional cloud group analysis diagram is shown in Figure 5a. Mode1 has a larger degree of redshift with a larger period, which is related to the excitation of the SPP mode. It can be found that the reflectivity of Mode 2 decreases significantly with the increase in the width of period P. The FWHM effectively shrinks. When p is 1280 nm, the reflectivity at the resonance wavelength of 1539.6 nm is as low as 0.09. The FWHN is 32.5 nm, nearly three times smaller than Mode 2 at the previous period of 1100 nm. To analyze the cause, we selected the reflection spectrum with a period p of 1280 nm. The electric field distribution of its corresponding Mode 2 is shown in Figure 5b. The electric field intensity of LSP of Mode 2, at the corners of upper and lower Au gratings, is obviously enhanced, and its electric field intensity even exceeds that of SPP mode.n.

Then, we made the same quantitative analysis on the grating width. When h1 = 120 nm, h2 = 80 nm, and *p* = 1100 nm, we scanned the grating width w in the range of 800 nm–1000 nm. The two-dimensional cloud group analysis diagram obtained after scanning is shown in Figure 6a. The reflection spectra of different widths in the scanning range are plotted, as shown in Figure 6b. With the increase in grating width, the resonant wavelength of Mode 1 changes less, which is related to the fact that Mode 1 is SPP mode. With the increasing grating width, the resonant wavelength of Mode 2 has a significant redshift, and the extinction ratio of its resonant dip also decreases. In Figure 6c,d, the electric field distribution diagrams of Mode 2, with grating widths w of 800 nm and 1000 nm, are displayed, respectively. When w is 800 nm, the electric field distribution is similar to Figure 5b. When w is 1000 nm, only the LSP at the corners of the upper and lower grating slits is obviously excited. The electric field intensity is significantly weaker than in Figure 6d.

Then, when h2 = 80 nm, *p* = 1100 nm, and w = 952 nm, we scanned the two-dimensional cloud group of reflectivity obtained by grating height h1 (100 nm–200 nm), as shown in Figure 7a. The change of Mode 1 is less, which is consistent with the previous analysis. Changing the grating height will not significantly impact its SPP mode. In contrast, the change in Mode 2 is larger. The extinction ratio of Mode 2 increases significantly. In Figure 7b,c the electric field distribution diagrams of Mode 2 with h1 of 100 nm and 200 nm are displayed, respectively. When h1 is 200 nm, the main reason for the larger extinction ratio is that the electric field intensity, at the corners of the upper Au grating, is significantly enhanced.

### 3.3. Dynamic Tuning of Fano Resonance by Laser-Induced Graphene(LIG)

Subsequently, we used a carbon dioxide laser to prepare laser-induced graphene to effectively heat the silicon wafer to regulate the Fano resonance measured in the experiment. The schematic diagram of the preparation is shown in Figure 8. The carbon dioxide laser burned the polyimide (PI) surface, causing lattice vibration due to the photo-thermal effect. The resulting high temperature destroys the C-O, C=O, and N-C bonds, and then, the remaining carbon atoms are reorganized to form laser-induced graphene (LIG) [24]. Laser-induced graphene has good thermal performance and is relatively stable, and its preparation cost is also very low. It can be prepared with only a piece of PI paper. Its shape and size can be customized. It is small and only has the thickness of ordinary paper, which is very suitable for metasurfaces. It is perfectly compatible with highly integrated metasurfaces and greatly saves the cost of preparing metal electrodes. Only a small drop of silver paste is needed to complete the preparation. Therefore, we attach a laser-induced graphene sheet under the metasurface device to control the heat generated by voltage, and the joule heat generated is transferred to silicon. Silicon has a high thermal-optical coefficient, effective regulation may be achieved in the Fano resonance.

With the increased voltage loaded on laser-induced graphene, Mode 1 and Mode 2’s resonant wavelengths have redshifts, as shown in Figure 9a. However, compared with Mode 2, the wavelength shift of Mode 1 can be neglected, as shown in Figure 9b. We locally amplified the electric field distribution, as shown in Figure 9a. The electric field distribution of Mode 1 is mainly concentrated in the air medium, while the hot spots formed by the LSP of Mode 2 are in silicon with a large thermo-optical coefficient. The Joule heat of laser-induced graphene can obviously make the resonant wavelength of Mode 2 shift to a longer wavelength. When 25 V is applied, the maximum redshift of the resonant wavelength of Mode 2 is 28.5 nm, while the shift of Mode 1 is only 3.6 nm. This shows the coupling of the two modes in Fano resonance can be controlled through the voltage.

## 4. Conclusions

In summary, we propose a new type of silicon-based Au grating metasurface structure, which uses the coupling of LSP and SPP, excited in the upper and lower Au gratings, to form the Fano resonance. The Fano resonance line type can be effectively changed by adjusting the parameters of the two-layer composite grating. The required Fano resonance can be obtained by designing different structural parameters. In the experiment, we realized the dynamic tuning of Fano resonance by using the Joule heat generated by laser-induced graphene. The coupling of two modes can be changed by adjusting only one mode in its Fano resonance. This scheme can be applied to the research of Fano resonance in optical communication, sensing, and optical nonlinearity.

## Figures and Tables

**Figure 1 nanomaterials-13-00492-f001:**
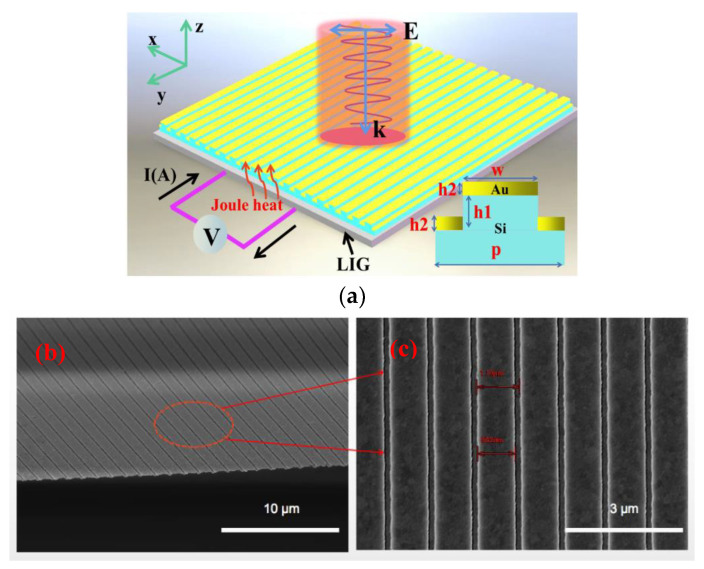
Design and fabrication of the structure (**a**); schematic diagram of device structure (**b**); oblique view of the scanning electron microscope (**c**); partial magnification of top view of the scanning electron microscope.

**Figure 2 nanomaterials-13-00492-f002:**
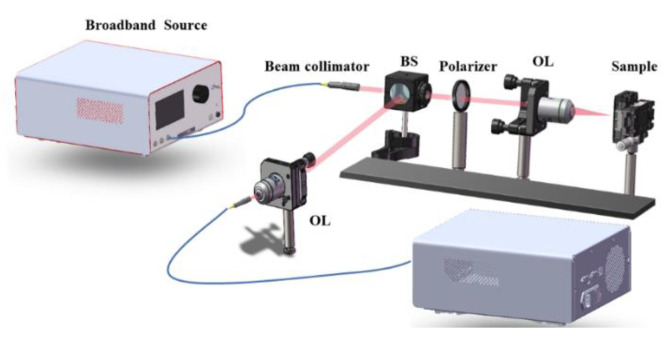
Schematic diagram of the experimental test setup.

**Figure 3 nanomaterials-13-00492-f003:**
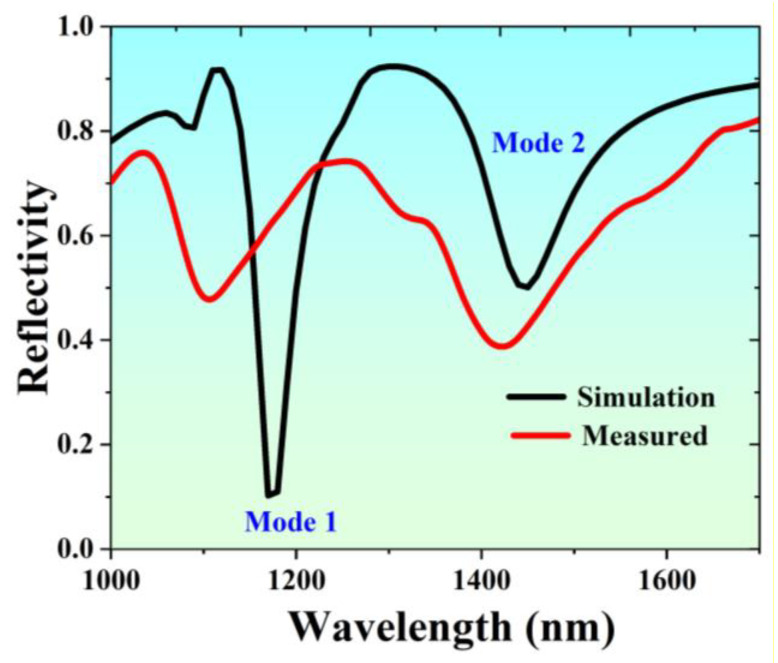
Reflectance spectra of devices tested by simulation and experiment, respectively.

**Figure 4 nanomaterials-13-00492-f004:**
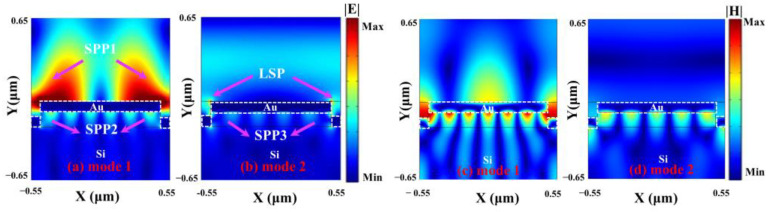
(**a**,**b**) Electric field distribution of Mode 1 and Mode 2, respectively. (**c**,**d**) Magnetic field distribution of Mode 1 and Mode 2, respectively.

**Figure 5 nanomaterials-13-00492-f005:**
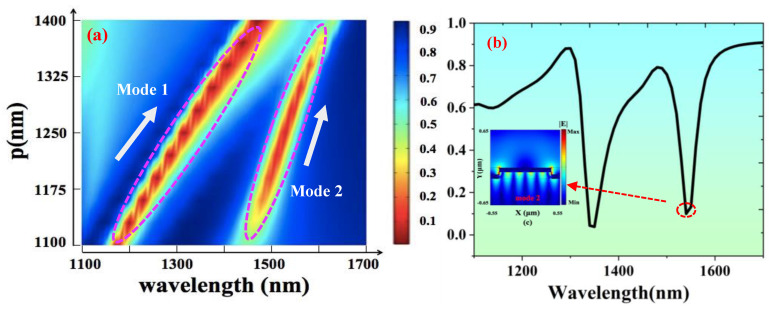
(**a**) A two-dimensional cloud group of the device refractive index obtained by scanning cycle width p (1100 nm–1400 nm) when h1 = 120 nm, h2 = 80 nm, and w = 952 nm. (**b**) Simulated reflection spectra at h1 = 120 nm, h2 = 80 nm, *p* = 1280 nm, and w = 952 nm. Inset: the normalized electric field distribution of the corresponding Mode 2.

**Figure 6 nanomaterials-13-00492-f006:**
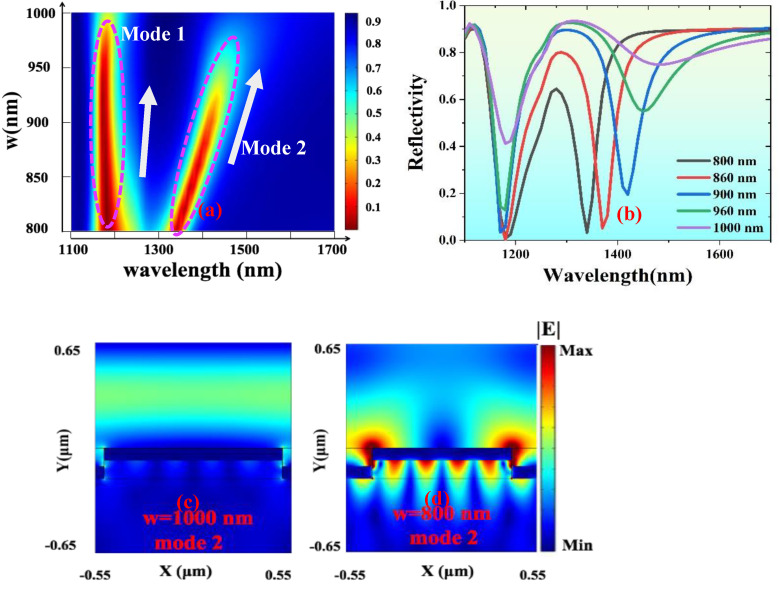
(**a**) When h1 = 120 nm, h2 = 80 nm, and *p* = 1100 nm, the reflectance of the device is in the variation range of 600 nm–800 nm for the grating width w. (**b**) Comparison of reflection spectra of devices at different grating widths. (**c**,**d**) Electric field distribution of Mode 2 when w is 800 nm and 1000 nm, respectively.

**Figure 7 nanomaterials-13-00492-f007:**
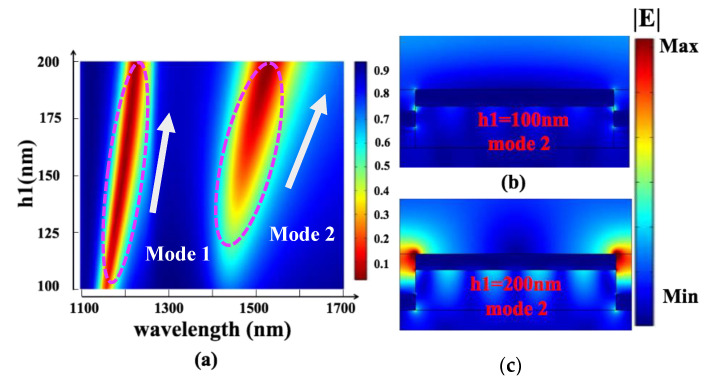
(**a**) The two-dimensional cloud group of device refractive index obtained by scanning the grating height h1 (100 nm–200 nm). (**b**,**c**) The electric field distribution of Mode 2 when h1 is 100 nm and 200 nm, respectively.

**Figure 8 nanomaterials-13-00492-f008:**
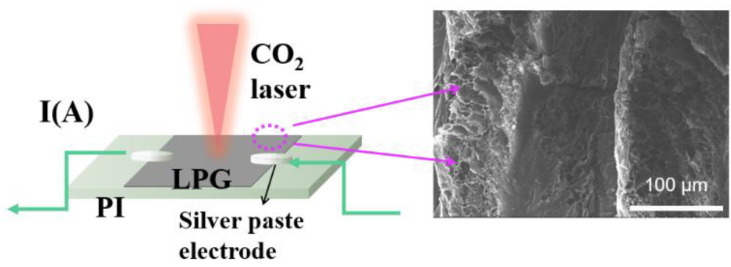
Schematic diagram of the preparation of laser-induced graphene, illustrated by the prepared scanning electron microscope. Inset: schematic diagram of electrically tunable.

**Figure 9 nanomaterials-13-00492-f009:**
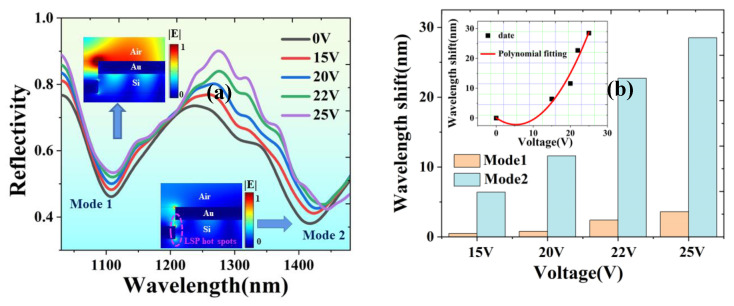
(**a**) Electrically tunable Fano resonance spectrum. Inset: local amplification of electric field distribution of Mode 1 and Mode 2. (**b**) Comparison of resonant wavelength shift between Mode 1 and Mode 2. Inset: Polynomialfitting of wavelength shift of Mode 2.

## Data Availability

The data presented in this study are available on request from the corresponding author.

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
