# Peer review of "Fano-Like Resonance of Heat-Reconfigurable Silicon Grating Metasurface Tuned by Laser-Induced Graphene"

_nanomaterials, 2023, doi:10.3390/nano13030492_

Round 1

Reviewer 1 Report

The manuscript presents novel results concerning light scattering from periodic nanogratings, however their interpretation and the overall presentation require revision of the manuscript according to comments listed below:

  1. Equations (2) and (3) require some explanation, what are the epsilons entering them? Moreover, just a sole mentioning of these equations is not sufficient: how exactly do they relate with the data presented? Do they explain the observed results quantitatively? 
  2. Fano-shaped resonances in 1D gratings discussed in this work are well known and extensively studied in the literature: [10.1103/PhysRevLett.91.183901; 10.1103/PhysRevB.70.125113; 10.1103/PhysRevB.76.201405; 10.1021/acsphotonics.8b01541]. In fact, they are known as “lattice plasmons” or “collective lattice resonances”, see relevant reviews: [10.1109/MAP.2015.2480083; 10.1021/acs.chemrev.8b00243; 10.1016/j.revip.2021.100051]. I highly encourage the authors to re-review the literature on this topic and put their work in the proper context.

Author Response

Please find the reply in the attachment.

Reviewer 2 Report

The paper provides experimental and numerical results the resonance peaks and their tuning by the electric current in the silicon/Au grating with the thin cover by the laser-induced graphene. The paper is good written and provides infesting results, but some important moments are still uncertain.

1. For example, the numerical and experimental results presented on Fig.3 are strongly differ for the case of the Mode 1 that has the dramatic discrepancy in the line width and the deep level. This fact needs to be explained.

2. The authors explains the numerical results of the decreasing the line width by the Fano resonance but I never see the Fano-like asymmetric resonance curve in the numerical and experimental results.

3. The resonance shift of the Mode 2 (see Fig.9) is explained by the local heating by electric current in the laser-induced grapheme. It will be good to make the additional prove the Joule heat nature of the wavelength shift by the numerical modeling. It look like that the parabolic fitting is more reasonable than linear one used in this case.

4. Besides, the electric current also provides the strong change in the reflection coefficient in the wide spectral range. What is the nature of these results? Does these results shown on Fig.9 are reproducing during the multiple cycling?

Thus the paper needs some improvement before the publication.

Author Response

(The authors gave the same response as above.)

Reviewer 3 Report

In this manuscript, the authors propose a tunable metasurface composed by a silicon grating coated with gold. The tuning is obtained thermally, by warming up the sample by applying a voltage.

While the work might be publishable in principle, there are several issues that need clarification because I can make a recommendation.

1.    The authors spend a huge fraction of the manuscript describing the optimization of the design, but very little text is devoted to describing the experiment and the tuning performance of the device. The authors show that the optical mode can be tuned by 28nm. However, this is less than the mode’s linewidth. Can the authors envision any potential application of this device? The authors mentioned ‘sensing’ in the manuscript. However, it is not clear to me how the demonstrated thermal tuning can improve the sensing capability of a device.

2.       The authors consider a device made of a silicon grating, with an ‘additional’ gold grating on top. However, it appears that the optical modes investigated by the authors are entirely plasmonic in their nature. Why did the author focus on plasmonic modes, which are typically very broad, instead of using on the lattice modes of the dielectric grating? It has been shown that Fano resonances in silicon gratings can be very sharp, and thus one would expect these modes to work better for sensing than plasmonic modes. Is it because plasmonic modes provide larger spectral tuning as the temperature increase? If so, why?

3.       Generally, the authors should add a more detailed discussion to compare their tuning approach to other approaches. How does this approach compare to, e.g., laser-induced heating?
See here https://nanoconvergencejournal.springeropen.com/articles/10.1186/s40580-019-0213-2  and here https://iopscience.iop.org/article/10.1088/2040-8986/abbb5b/pdf for two review papers summarizing recent results of tunable metasurfaces.  See also https://arxiv.org/abs/2210.05586 for a recent work where thermo-optic effects in silicon metasurfaces are used to induce a frequency shift comparable or larger than the linewidth.

4.       Connected to previous point: can the author provide an estimation of the temperature shift in their sample?

5.       I am a bit confused by Fig. 9 and its description in the text. Is Fig. 9a an experiment or simulation? In the text, the authors say “We locally amplified the electric field distribution of the device and reselected the maximum and minimum values of electric field intensity, as shown in Figure 9 (a).” I am not sure about the meaning of “locally amplified the electric field distribution”, and whether it describes an experimental procedure or a numerical calculation.

6.       Can the authors clarify how and where they apply voltage to their sample (for the experiment in Fig. 9)? Are both electrodes on the LIG? How much current is flowing? And is the spectral tuning reversible?

7.       The caption of Fig. 6 is written in a weird way. The first 3 rows of the caption sound like a set of instructions.

Author Response

(The authors gave the same response as above.)

Round 2

Reviewer 3 Report

The authors have addressed all my concerns, the manuscript can be published.